# Epidemiological Study of Malignant Paediatric Liver Tumours in Denmark 1985–2020

**DOI:** 10.3390/cancers15133355

**Published:** 2023-06-26

**Authors:** Thomas N. Nissen, Catherine Rechnitzer, Birgitte K. Albertsen, Lotte Borgwardt, Vibeke B. Christensen, Eva Fallentin, Henrik Hasle, Lars S. Johansen, Lisa L. Maroun, Karin B. Nissen, Allan Rasmussen, Mathias Rathe, Steen Rosthøj, Nicolai A. Schultz, Peder S. Wehner, Marianne H. Jørgensen, Jesper Brok

**Affiliations:** 1Department of Paediatrics and Adolescent Medicine, Rigshospitalet, Copenhagen University Hospital, 2100 Copenhagen, Denmark; catherine.rechnitzer@regionh.dk (C.R.); vibeke.brix.christensen@regionh.dk (V.B.C.); marianne.hoerby-joergensen@regionh.dk (M.H.J.); jesper.sune.brok@regionh.dk (J.B.); 2Department of Paediatrics and Adolescent Medicine, Aarhus University Hospital, 8200 Aarhus, Denmark; biralber@rm.dk (B.K.A.); henrik.hasle@skejby.rm.dk (H.H.); karins@rm.dk (K.B.N.); 3Department of Clinical Medicine, Faculty of Health, Aarhus University, 8200 Aarhus, Denmark; 4Department of Radiology, Rigshospitalet, Copenhagen University Hospital, 2100 Copenhagen, Denmark; lotte.borgwardt@regionh.dk (L.B.); eva.fallentin@regionh.dk (E.F.); 5Department of Paediatric Surgery, Rigshospitalet, Copenhagen University Hospital, 2100 Copenhagen, Denmark; lars.soendergaard.johansen@regionh.dk; 6Department of Pathology, Rigshospitalet, Copenhagen University Hospital, 2100 Copenhagen, Denmark; lisa.leth.maroun@regionh.dk; 7Department of Surgery and Transplantation, Rigshospitalet, Copenhagen University Hospital, 2100 Copenhagen, Denmark; allan.rasmussen@regionh.dk (A.R.); nicolai.aagaard.schultz.01@regionh.dk (N.A.S.); 8Department of Paediatric Haematology and Oncology, H. C. Andersen Children’s Hospital, Odense University Hospital, 5000 Odense, Denmark; mathias.rathe@rsyd.dk (M.R.); peder.skov.wehner@rsyd.dk (P.S.W.); 9Department of Paediatrics and Adolescent Medicine, Aalborg University Hospital, 9000 Aalborg, Denmark; steen.rosthoej@rn.dk

**Keywords:** liver tumours, hepatoblastoma, paediatric, epidemiology

## Abstract

**Simple Summary:**

Malignant liver tumours in children are rare and outcomes seldom published. A global collaboration has recently been initiated to ensure improved treatment, and, in this study, we report the incidence, outcomes and long-term adverse events in a Danish population during the last 35 years. We included 79 patients in the analysis, and the overall incidence was ~2.29 per 1 million children (<15 yr) per year. Hepatoblastoma was the most frequent tumour, with 61 cases. Overall survival 5 years after diagnosis was 84% and 78% for hepatoblastomas and hepatocellular carcinomas, respectively. Age ≥ 8 years was the most predominant factor associated with a poorer outcome. Adverse events included reduced renal function in 10%, reduced cardiac function in 6% and impaired hearing function in 60%, where 19% needed hearing aids. Survival after hepatoblastoma in childhood has improved during the last 35 years and is comparable with international results.

**Abstract:**

Background: Malignant liver tumours in children are rare and national outcomes for this tumour entity are rarely published. This study mapped paediatric liver tumours in Denmark over 35 years and reported on the incidence, outcomes and long-term adverse events. Methods: We identified all liver tumours from the Danish Childhood Cancer Registry and reviewed the case records for patient and tumour characteristics, treatment and clinical outcome. Results: We included 79 patients in the analyses. Overall crude incidence was ~2.29 per 1 million children (<15 yr) per year, with 61 hepatoblastomas (HB), 9 hepatocellular carcinomas and 9 other hepatic tumours. Overall 5-year survival was 84%, 78% and 44%, respectively. Nine patients had underlying liver disease or predisposition syndrome. Seventeen children underwent liver transplantation, with two late complications, biliary stenosis and liver fibrosis. For HB, age ≥ 8 years and diagnosis prior to 2000 were significant predictors of a poorer outcome. Adverse events included reduced renal function in 10%, reduced cardiac function in 6% and impaired hearing function in 60% (19% needed hearing aids). Behavioural conditions requiring additional support in school were registered in 10 children. Conclusions: In Denmark, incidences of malignant liver tumours during the last four decades have been increasing, as reported in the literature. HB survival has improved since the year 2000 and is comparable with international results. Reduced hearing is the major treatment-related side effect and affects approximately 60% of patients.

## 1. Introduction

In children, malignant tumours of the liver are rare and represent approximately 2% of all malignant paediatric tumours. The most frequent are hepatoblastomas (HB), affecting very young children; less frequent are hepatocellular carcinomas (HCC), in older children or adolescents. Typically, liver tumours present as an abdominal mass in the upper right quadrant in an unwell child and alpha fetoprotein-1 (AFP) is elevated in most HB [1]. The incidence varies slightly according to the region, and known risk factors are hepatitis B infection, underlying cirrhosis, tyrosinemia, cholestatic liver disease, low birth weight and several predisposition syndromes, e.g., Beckwith-Wiedemann syndrome (BWS) [2,3,4,5]. The incidence of HB in the US was 1.4 per million year (1992–2004), but studies have also reported increasing incidences since 1970s [6,7,8,9]. The overall 5-year survival of HB varies from 82 to 88% if optimal treatment is available [7,10,11,12].

Because hepatic tumours are rare, the four major collaborative groups (the International Childhood Liver Tumour Strategy Group (SIOPEL), the Children’s Oncology Group (COG), the German Society for Paediatric Oncology and Haematology and the Japanese Study Group for Paediatric Liver Tumours) have founded an international collaboration (the Children’s Hepatic tumours International Collaboration (CHIC)), to create a common approach to risk stratification and treatment [13,14,15].

Treatment of HB and HCC has evolved from solely upfront surgery to predominantly cisplatin-based chemotherapy with delayed surgery and ultimately liver transplantation in challenging cases or tumours involving all liver segments.

This has reduced the risk of relapse and thus improved overall survival [16,17,18]. The dominating long-term adverse event following a cisplatin-based approach is ototoxicity with the risk of irreversible hearing loss. This remains a major challenge, although the introduction of sodium thiosulfate has proven to reduce toxicity without affecting overall survival [19,20,21,22]. Although less frequent, reduced renal and cardiac function are other known side effects related to chemotherapy.

In Denmark, the treatment of liver tumours has followed international protocols from SIOPEL since the 1990s [18]. The administration of chemotherapy is applied in a collaborative setting at four paediatric oncology departments in the country, and surgery is centralised at one institution [23]. Since 2019, treatment has followed the protocol from the Paediatric Hepatic International Tumour Trial (PHITT) [24].

In this study, we report on the overall survival of patients with malignant paediatric liver tumours in Denmark during the last 35 years. We also analyse risk factors and treatment-related adverse events.

## 2. Materials and Methods

From the Danish Childhood Cancer Registry [25], we identified all malignant liver tumours (ICD10: DC22) in children aged 0–14 years and diagnosed between 1985 and 2020. The case record of each patient was reviewed (TN, CR and JB) for patient and tumour characteristics, treatment, adverse events and clinical outcomes, including long-term data on liver, cardiac, kidney and hearing function, along with growth and school performance. The Danish Childhood Cancer Registry is linked to the patient’s unique personal identification number and allows an accurate link to the case record. 

To assess factors associated with survival, the following risk factors were defined a priori: sex, year of diagnosis (</≥ year 2000), liver transplantation, PRETEXT groups (I–III and IV), age at diagnosis (</≥8 years), alpha fetoprotein-1 </≥ 200 µg/L and lung metastases. The latter four factors are in alignment with the risk stratification in the PHITT protocol. Survival curves were calculated by the Kaplan–Meier method and compared by Mantel–Cox log-rank test. *p*-values below 0.05 were considered statistically significant.

From Statistics Denmark (https://www.statistikbanken.dk/FAM111N, accessed on 1 December 2022), annual age-specific childhood population data were used for the calculation of all incidences. All incidences are reported as events per million years. Incidences were also calculated in three age groups, 0–4 years, 5–9 years and 10–14 years, and in three time periods, 1985–1996, 1997–2008 and 2009–2020, to assess the incidence’s development over time.

All statistical analysis were performed using STATA 13.1 (Stata Corp LP, College Station, TX, USA) and all figures were created using GraphPad Prism 9.4.1 (GraphPad Software, Inc., San Diego, CA, USA).

## 3. Results

A total of 84 paediatric patients with malignant liver tumours were identified in Denmark from 1985 to 2020. Detailed medical records were unavailable in five patients from the 1980s, who were excluded from the analysis. Thus, 79 patients were included. Tumours were diagnosed as HB in 61 patients, HCC in 9 and other tumours in 9 (embryonal rhabdomyosarcoma, hemangiosarcoma, mesothelioma, epithelial tumour). Nine children (four HB, three HCC and two other tumours) had an underlying liver disease, e.g., congenital cirrhosis, cholestatic liver disease or tyrosinemia, or a predisposition such as BWS or familial adenomatous polyposis. Six of nine children with HCC had no underlying liver disease. 

Children with HB were diagnosed at a median age of 1.7 years (IQR 0.96–2.5 years) and 57% were boys. Four of the children with HB were born premature (5%), with birthweights from 655 to 974 g and gestational ages (GA) from 26 to 29 weeks. All tumours were histologically classified as mixed epithelial or mixed epithelial and mesenchymal. Non-tumoural-adjacent liver tissue showed cirrhosis in four cases (one HB, two HCC and one other tumour) and cholestatic disease in one HCC. None of the tumours were hepatitis-B-related (Table 1). HCC was characterised by a higher median age of 10.6 years (IQR 6.9–13.8 years) at diagnosis, whereas other liver tumours were diagnosed at ages comparable with HB (Appendix A). The treatment of all children but two, treated prior to 1990, followed the SIOPEL 1–6 protocols.

### 3.1. Incidence of All Liver Tumours and Hepatoblastoma

The overall incidence for any malignant liver tumour was 2.29 cases per million year. The incidence was significantly higher among the youngest children (<5 yr). Dividing the study cohort into three periods, we observed a statistically non-significant tendency towards increasing incidence, from 1.76 per million year in the first period to 2.82 per million year in the most recent years. Likewise, an increasing incidence was seen in every age group (Table 2).

For HB, the overall incidence was 1.77 cases per million year and it occurred predominantly among children <5 years old. There was also a tendency towards an increasing incidence (Table 3).

### 3.2. Tumour and Treatment Characteristics of Hepatoblastoma

At diagnosis, children with HB had a median AFP of 180,500 µg/L (IQR 14,200–530,000 µg/L). Lung metastases were found in 14 children (22%), extrahepatic vascular involvement in 16 children (26%) and local invasive spread in six children (10%). Thirteen patients were stratified as PRETEXT group IV (22%), which also included five with lung metastases. Pre-operative chemotherapy was given to all but three children who had upfront surgery, with cisplatin (26%), cisplatin and doxorubicin (31%) or cisplatin, doxorubicin and carboplatin (33%). Four children had no available chemotherapy data. After surgery, 57% of the children received adjuvant chemotherapy with cisplatin (25%), cisplatin and doxorubicin (21%) or cisplatin, doxorubicin and carboplatin (12%); no chemotherapy was given following liver transplantation and in two patients following a complete tumour response (25%).

Surgery was performed in all but five children (8%) who had inoperable tumours and one child (2%) who had no visible tumour following a single round of chemotherapy. Surgery consisted of LTX in 9 children (15%), minor liver resection (<3 segments) in 13 (21%), hemi-hepatectomy in 25 (41%) and extended hemi-hepatectomy in 7 (12%)(Table 1).

### 3.3. Overall and Event-Free Survival of Hepatoblastoma

Overall 5-year survival was 79% for any liver tumour with different subtypes, compared to the survival of patients with HB (84%), HCC (78%) and other tumours (44%). HB had a better survival rate compared to HCC, with hazard ratio 1.26 (CI: 0.28–5.77), and compared to other liver tumours, with hazard ratio 5.06 (CI: 1.71–14.9). A *p*-value of 0.005 was used with the Mantel–Cox log-rank test for differences (Figure 1).

Overall, 5-year event free survival was 76% with the different subtypes, HB (82%), HCC (67%) and other tumours (44%). Survival analysis using the Cox regression model showed statistically significantly higher event-free survival in children with HB compared to HCC, with hazard ratio 1.79 (CI: 0.50–6.44), and to other liver tumours, with hazard ratio 4.79 (CI: 1.64–13.89). A *p*-value of 0.007 was used with the Mantel–Cox log-rank test for differences (Figure 1).

### 3.4. Causes of Death

After 1985, 21 deaths occurred: 13 among HB (10 before 5 years, 1 before 10 years and 2 later than 10 years after diagnosis), 3 among HCC and 5 among other tumours. Four patients with HB died of complications: liver vein thrombosis and rescue LTX, postoperative sepsis or following very late LTX complications after 16 and 21 years, respectively. The other 17 patients died of disease progression.

### 3.5. Risk Factor Analysis of 5-Year Survival

Age at diagnosis ≥8 years was a significant risk factor for mortality, with an HR of 6.26 (CI: 1.76–22.24, *p*-value: 0.005) and 6.13 (CI: 1.78–21.17, *p*-value: 0.004) for overall survival and event-free survival, respectively. For the other pre-defined known risk stratification factors (PRETEXT, lung metastases and LTX), there was a trend that all had an impact on overall survival (*p*-value 0.10, 0.17 and 0.28) and also event-free survival but it was not statistically significant; see Table 4.

To test the cutoff of 8 years of age, we created three different spline regression models to predict the ages with the highest risk of mortality. Analysis indicated that the risk of mortality was highest around 8 years of age, but that a biphasic curve was present with increasing mortality around 2 years of age (Table 4 and Figure 2 and Appendix A).

AFP was not analysed as only one patient had AFP lower than the cutoff value of 200 µg/L.

To assess the outcomes of treatment over time during the study period, we compared survival data before and after the year 2000. This showed a lower mortality risk after the year 2000, with an HR of 0.20 (CI: 0.05–0.79, *p*-value: 0.02) and 0.27 (CI: 0.08–0.92, *p*-value: 0.04) for overall survival and event-free survival, respectively (Table 4).

### 3.6. Health Condition after Treatment for Hepatoblastoma

All survivors of liver tumours were followed for long-term treatment complications.

The nine survivors post-LTX (6 HB, 3 HCC) were followed for up to 22 years and other survivors for up to 32 years.

No case of retarded growth or overweight was registered in either of the two groups.

Liver complications were diagnosed in three cases: one following surgery, developing portal vein occlusion and splenomegaly, and two after LTX, with biliary stenosis and de novo autoimmune hepatitis in one patient and liver fibrosis in another. Liver function biochemistry was without significant abnormalities.

Marginally reduced kidney function (GFR < 90 mL/min/1.73 m^2^) was seen in four children. However, one patient had severe chronic hypomagnesemia and hypophosphatemia due to cisplatin-related tubular damage.

No patient showed clinical symptoms of cardiac insufficiency. Only four children (6%) had borderline EF ≤ 55%, but none of the patients had an EF below 50%.

Impaired hearing function with Boston score >1 was seen in 60 % of the children. Moreover, 33% of patients with Boston 3 or 4 and nine patients (19%) needing hearing aids (Table 5).

We calculated the cumulative dosage of chemotherapy during treatment. The mean cumulative dosage was 250 mg/m^2^, 452 mg/m^2^ and 621 mg/m^2^ for doxorubicin, cisplatin and carboplatin, respectively. No differences in cumulative cisplatin dosage were found between children with and without hearing sequelae after treatment; likewise, no association was found for cumulative doxorubicin and reduced EF.

The assessment of school performance revealed that developmental disorders (attention deficit hyperactivity disorder (ADHD), attention deficit disorder (ADD), obsessive-compulsive disorder (OCD) and autism) requiring additional support in school were registered in 10 children (eight boys and two girls). The other 51 children had school performance within the normal range.

## 4. Discussion

This study reports thoroughly, via the unique Danish identification number, on all liver malignancies in Danish children during a 35-year period (1985–2020). Throughout the study period, children with HB and HCC in Denmark were treated according to the SIOPEL guidelines at four institutions; however, surgery has been centralised at one institution.

### 4.1. Main Findings

In this study of 79 children with liver malignancies, we found, as expected, that HB was the most frequent, with 77% of the cases. HB was diagnosed at a median age of 608 days. Overall incidence throughout the study period was 2.29 cases per million year and there was a tendency towards an increasing incidence over time. Lung metastasis was found in 25% of the children and 22% were stratified as PRETEXT group IV. Other liver malignancies were extremely rare, with HCC being the second most frequent.

Overall, the 5-year survival of HB was 84% and was significantly better than any other liver tumour. Risk factors (PRETEXT, lung metastases and LTX) related to tumour burden showed a clear tendency towards higher mortality, but our sample size was likely too small to obtain statistical significance. Age ≥ 8 years was the clearest predictor of an increased risk of mortality, with an HR of 6.26 (*p*-value: 0.005). Children diagnosed after the year 2000 had better outcomes, indicating that treatment has been improving compared to the first study period (HR: 0.20, *p*-value: 0.02)

### 4.2. Comparison with Other Studies

Publications on national studies on liver tumours are scarce, probably due the rarity of the tumour entity [23]. An assessment of HB in all the Nordic countries (Denmark, Sweden, Norway and Finland) from 1985 to 2006 reported an overall incidence of 1.7 cases per million year, which is comparable to the incidence found in the present study [9]. Surprisingly, a recent Finnish study found only an incidence of 0.66 cases per million in a 30-year period from 1987 to 2017, which was also noted as lower than other European reports [11].

In the Finnish study, the overall survival of HB was 86%, and a Swedish study from 1983 to 2007 found overall survival of 68% in the first half of the study period (1983–1995) and 88% in the second half of the study period (1995–2007) [10]. These data suggest that overall survival in Danish children with HB is comparable to that of other Nordic countries.

In Japan, a study of HB between 1999 and 2012 reported an overall survival rate of 83%, also in line with our findings. Age at diagnosis > 3 years (HR 3.25) and lung metastases (1.52) were statistical significant predictors of an increased risk of mortality [12]. In our study only, age > 8 years was a statistically significant predictor, with an HR of 2.90 and 2.43 for PRETEXT IV and lung metastases, respectively, which may indicate an association similar to that in the Japanese study.

A US analysis of 511 cases of HB between 2004 and 2015 indicated an increasing incidence in children 0–18 years. In 2000, the incidence of 1.89 cases per million year increased to 2.16 cases per million year in 2016, where we found an incidence in 1985–1996 of 1.30 cases per million year, increasing to 2.14 cases per million year in 2009–2020. They found an overall survival rate of 82%, very similar to all other data, e.g., in other Nordic countries and Japan [7]. Overall, there is a trend worldwide of a slightly increasing incidence of HB but accompanied by improved overall survival.

In our study, we found few survivors with reduced cardiac output or kidney function, and, if affected, the clinical impact was usually modest. This mirrors the results of a Nordic cohort study that assessed hospitalisation in HB survivors with a median follow-up time of 11 years. In 86 patients, they reported no events of hospitalisation for chronic kidney disease or cardiac diseases [26]. In contrast, our study revealed that over half of the children treated for HB suffered from impaired hearing (Boston score > 1) and one third from severe hearing loss (Boston score 3 + 4). The is in line with previous studies, and the dosage of cisplatin has been an important focus in recent trial designs [19,20,22,27,28]. Studies have shown that cisplatin-induced hearing loss in paediatric patients usually occurs following cumulative doses exceeding 400 mg/m^2^ [27,29,30,31]. This is also seen among other paediatric tumours, e.g., neuroblastomas and brain tumours [27]. In contrast, our study indicated that the children with impaired hearing function received comparable cumulative doses of cisplatin to children with normal hearing function, but our sample size was small. The children included in our study were treated at a very young age. Thirty-seven (61%) were below 2 years of age. This could explain the relatively high occurrence of ototoxicity that we observed.

Only nine (15%) children with HB had LTX as primary surgery. An HR of 2.10 compared to tumour resection could indicate an increased risk of mortality, but the numbers are small. This can be compared to the Finish study, where a total of 19 patients (40%) received LTX as a primary or rescue surgery, with an OS at 76% and 68%, respectively. Noteworthy is that 12 (63%) of the Finnish patients had post-transplant complications, against four (36%) in our study. Larger assessments of US data as well as European studies indicate that LTX is associated with the same overall survival rates [30,31,32,33,34,35].

### 4.3. Strengths and Limitations

The main strength of this study is that the Danish Childhood Cancer Registry is linked to the Danish social security number (CPR) and all children with malignancies are registered. Thus, thorough data are retrievable, and it is possible to include all children with a malignancy. The study was a collaboration with all national paediatric oncologists and specialists in surgery, radiology, pathology and hepatology. Thereby, all relevant clinical inputs were collected, which further ensured the completeness of our data.

The main limitation of the study was the retrospective design and the relatively small sample size, as Denmark has a population of approximately 5.8 million inhabitants. A larger sample size is needed to reach statistical significance, e.g., in terms of PRETEXT and lung metastases as significant prognostic factors. This stresses the need to include international protocols for rare diseases.

## 5. Conclusions

In conclusion, the incidence of hepatic tumours in Denmark is comparable to that in other international publications and is also showing a trend of increasing incidence. Survival for HB in Denmark is improving over time and is comparable to international results. Prognostic and treatment-stratifying factors in current international protocols were supported in the Danish cohort. Hearing loss was the major treatment-related side effect and affected approximately 60% of the children.

## Figures and Tables

**Figure 1 cancers-15-03355-f001:**
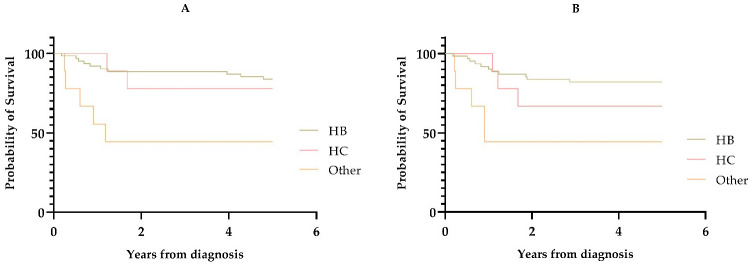
Kaplan–Meier curves of survival 5 years after diagnosis. HB: hepatoblastoma, HCC: hepatocellular carcinoma. (**A**) Overall 5-year survival. (**B**) Event-free 5-year survival.

**Figure 2 cancers-15-03355-f002:**
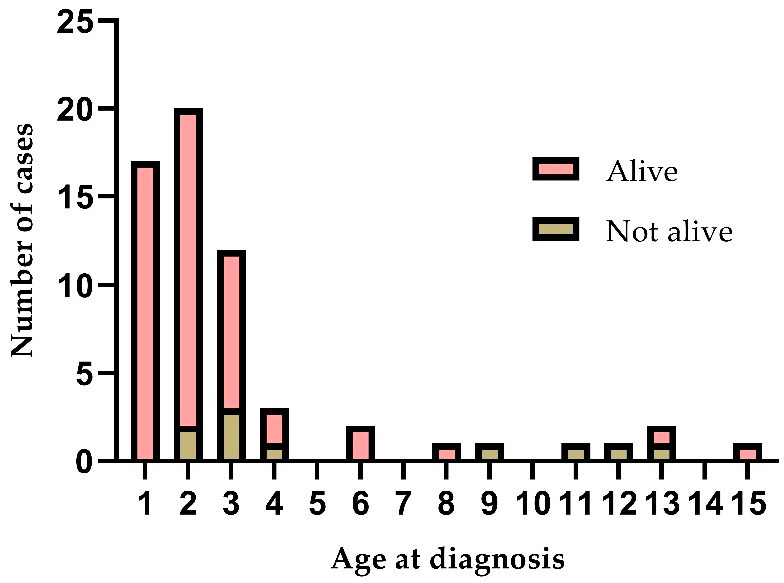
Histogram of all children diagnosed with hepatoblastoma divided by living status 5 years after diagnosis.

**Table 1 cancers-15-03355-t001:** Characteristics of study participants.

	All Liver Tumours N = 79	HB N = 61
	*n* (%)	*n* (%)
Sex: male	43 (54)	35 (57)
Age at diagnosis (years, median IQR)	1.96 (1.0–5.6)	1.7 (0.96–2.5)
Alpha fetoprotein-1 µg/L (median, IQR)		180,500 (14,200–530,000)
PRETEXT group IV		13 (22)
Extrahepatic vessels	18 (23)	16 (26)
Local nodes	8 (10)	6 (10)
Distant metastases	19 (24)	15 (25)
Pre-surgery chemotherapy		
Cisplatin	16 (20)	16 (26)
Cisplatin/Doxorubicin	20 (25)	19 (31)
Cisplatin/Doxorubicin/Carboplatin	24 (30)	20 (33)
None	8 (10)	3 (5)
NA	5 (6)	3 (5)
Other	6 (8)	
Type of primary surgery		
LTX	14 (18)	9 (15)
Minor liver resection (<3 segments)	14 (18)	13 (21)
Hemi-hepatectomy	29 (37)	25 (41)
Extended hemi-hepatectomy	8 (10)	7 (12)
Inoperable	10 (13)	5 (8)
No tumour	2 (3)	1 (2)
NA	2 (3)	1 (2)
Rescue LTX	3 (4)	2 (3)
Post-surgery chemotherapy		
Cisplatin	15 (19)	15 (25)
Cisplatin/Doxorubicin	13 (17)	13 (21)
Cisplatin/Doxorubicin/Carboplatin	8 (10)	7 (12)
None	10 (12)	9 (15)
Other	33 (42)	17 (28)
Relapse	9 (11)	6 (10)

HB: hepatoblastoma, NA: not available, LTX: liver transplantation.

**Table 2 cancers-15-03355-t002:** Incidence of all liver tumours ^1^.

Age at Diagnosis	1985–1996	1997–2008	2009–2020	1985–2020
<5 yr	4.50	4.23	6.72	5.14
5–10 yr	0.29	0.98	0.77	0.70
>10 yr	0.52	1.56	1.23	1.11
All patients	1.76	2.26	2.82	2.29

^1^ All incidences are reported as event per million year.

**Table 3 cancers-15-03355-t003:** Incidence of hepatoblastoma ^1^.

Age at Diagnosis	1985–1996	1997–2008	2009–2020	1985–2020
<5 yr	3.94	4.23	5.65	4.61
5–10 yr	0.00	0.74	0.26	0.35
>10 yr	0.00	0.52	0.74	0.43
All patients	1.30	1.85	2.14	1.77

^1^ All incidences are reported as event per million year.

**Table 4 cancers-15-03355-t004:** Survival analysis of hepatoblastoma after 5 years of follow-up.

	Overall Mortality Risk		Overall Event Risk	
	HR (95% CI) ^1^	*p*-Value ^2^	HR (95% CI) ^1^	*p*-Value
Sex				
Male	1		1	
Female	0.90 (0.25–3.18)	0.87	1.11 (0.34–3.65)	0.85
PRETEXT				
I–III	1		1	
IV	2.90 (0.82–10.30)	0.10	2.50 (0.73–8.55)	0.15
Age at diagnosis				
<8 years	1		1	
≥8 years	6.26 (1.76–22.24)	0.005	6.13 (1.78–21.17)	0.004
LTX				
no	1		1	
yes	2.10 (0.54–8.13)	0.28	1.99 (0.53–7.49)	0.31
Lung metastases				
No	1		1	
Yes	2.43 (0.68–8.64)	0.17	1.97 (0.58–6.74)	0.28
Year of diagnosis				
Before year 2000	1		1	
After year 2000	0.20 (0.05–0.79)	0.02	0.27(0.08–0.92)	0.04

HR: hazard ratio, ci: confidence interval, ltx: liver transplantation. ^1^ HR and CI calculated from Cox regression models. ^2^
*p*-values calculated from Mantel–Cox log-rank test.

**Table 5 cancers-15-03355-t005:** Sequelae after treatment of hepatoblastoma.

	HB (N = 48) N (%)
ECHO EF < 55	3 (6)
Boston score > 1	29 (60)
Boston 3 + 4	16 (33)
Hearing aids	9 (19)
GFR < 90 mL/min/1.73 m^2^	5 (10)

HB: hepatoblastoma, EF: ejection fraction, GFR: glomerular filtration rate. All estimates calculated as number needed to treat.

## Data Availability

Data are available via the authors from a local database if required or needed.

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
