# Peer review of "Epidemiological Study of Malignant Paediatric Liver Tumours in Denmark 1985–2020"

_cancers, 2023, doi:10.3390/cancers15133355_

Round 1

Reviewer 1 Report

This is a descriptive epidemiologic study summarizing  incidence, outcomes and long-term adverse events in Denmark overs a period of 35 years.  Hepatoblastomas, hepatocellular carcinoma and other liver tumors  were included in the analysis.  

One additional parameter that should be included in the study is the histology of the adjacent non-tumor liver tissue, (normal, non-cirrhotic, cirrhotic).  If this information is available it would be of interest to include in all cases (particularly in the HB cases with age>4,  and in the HCC cases).

Also, no HCC cases were Hepatitis B related? (Was serologic HBV data available in all HCC cases?)

Lines 120-123, the authors state that  9 patients had underlying liver disease; it would be helpful to include a table with each case associated with underlying liver disease (patient age at diagnosis,  patient sex , tumor type, underlying liver disease, histology of non-tumor liver (normal, non-cirrhotic, cirrhotic).

If available for HCC, it would be helpful to document the histologic grade of the tumor. If this information is available, assess overall and event free survival for low grade vs high grade  (well differentiated vs moderately/poorly differentiated) HCC

Since the total group of patients is quite small (79 patients), a supplementary summary table of data for the entire group patients including the following parameters (age of diagnosis, sex, tumor diagnosis, tumor stage, treatment, overall survival) would be helpful to include. 

In the current paper, suppl table... some minor  (English spelling ) corrections - "tabel"  (table)

-"Metasis " (Metastasis)

-"Relaps" (Relapse)

Author Response

Dear reviewer 1,

Thank you for your comments and giving us the possibility to improve the manuscript. Below we have addressed all your comments – written in italic.

This is a descriptive epidemiologic study summarizing  incidence, outcomes and long-term adverse events in Denmark overs a period of 35 years.  Hepatoblastomas, hepatocellular carcinoma and other liver tumors  were included in the analysis.  

One additional parameter that should be included in the study is the histology of the adjacent non-tumor liver tissue, (normal, non-cirrhotic, cirrhotic).  If this information is available it would be of interest to include in all cases (particularly in the HB cases with age>4,  and in the HCC cases).

This is added: Non-tumoral adjacent liver tissue showed cirrhosis in 4 cases (I HB, 2 HCC and 1 other tumor) and intrahepatic cholestasis in one HCC. Liver tissue was normal in all other cases.

Also, no HCC cases were Hepatitis B related? (Was serologic HBV data available in all HCC cases?)

Serology for HBV was available in all patients. None of the tumors was hepatitis B related.

Lines 120-123, the authors state that  9 patients had underlying liver disease; it would be helpful to include a table with each case associated with underlying liver disease (patient age at diagnosis,  patient sex , tumor type, underlying liver disease, histology of non-tumor liver (normal, non-cirrhotic, cirrhotic).

Only 5 patients had an underlying liver disease, 3 are HCC. We wrote that “6 of 9 HCC had no underlying disease”. The 4 other patients had a predisposition. We consider that a table would not add to the analysis of outcome.

If available for HCC, it would be helpful to document the histologic grade of the tumor. If this information is available, assess overall and event free survival for low grade vs high grade  (well differentiated vs moderately/poorly differentiated) HCC.

This information is not available in our database.

Since the total group of patients is quite small (79 patients), a supplementary summary table of data for the entire group patients including the following parameters (age of diagnosis, sex, tumor diagnosis, tumor stage, treatment, overall survival) would be helpful to include. 

In table 1, the first column presents data of the entire group.

In the current paper, suppl table... some minor  (English spelling ) corrections - "tabel"  (table) Corrected

-"Metasis " (Metastasis) Corrected

-"Relaps" (Relapse) Corrected

Reviewer 2 Report

This study primarily describes the incidence, treatment outcomes, and long-term adverse reactions of malignant liver tumors in Danish children. The research analyzed data from 79 patients, with hepatoblastoma being the most common type of liver tumor. Overall, the incidence of malignant liver tumors in children was approximately 2.29 per million children, with a 5-year survival rate ranging from 84% to 78%. Age over 8 years was identified as the most significant factor affecting treatment outcomes. In terms of adverse reactions, 10% of patients experienced renal dysfunction, 6% experienced cardiac dysfunction, 60% experienced hearing impairment, and 19% required hearing aids. Overall, the treatment outcomes of hepatoblastoma in Danish children have improved and are comparable to international standards.

In fact, the incidence of malignant liver tumors in children, including in China, has been increasing year by year. This is partly due to improved diagnostic capabilities and the influence of factors such as food and the environment. Various treatment methods, including chemotherapy, as well as the use of multidisciplinary teams (MDT), have improved the effectiveness of treatment for malignant liver tumors in children. However, corresponding complications, such as renal toxicity and hearing impairment, are issues that clinicians must pay attention to. This article reveals the epidemiological characteristics of childhood malignant liver tumors in Denmark over the past few decades, which has certain clinical significance. The reviewer overall believes that revisions are necessary but maintains a positive attitude towards publication.

There are several issues with this article, as follows:

1.     In the introduction section, the third paragraph lacks content. The author is requested to provide additional information.

2.     The formatting of Table 1 needs to be standardized. It may be better to add parentheses for the quantity annotation in the first column.

3.     The article includes statistical analysis of the incidence of hepatoblastoma and malignant liver tumors in different years (1985-1996, 1997-2008, 2009-2020, and 1985-2020). It may be worthwhile to further analyze and compare the long-term survival rates of hepatoblastoma and malignant liver tumors in these different years, providing more clinical guidance.

4.     Since age over 8 years is an important factor affecting prognosis, it would be beneficial to compare the cardiac, renal, and hearing complications of hepatoblastoma patients over 8 years old with those of the overall population of children with malignant liver tumors. The author can further conduct statistical analysis to identify any differences in risk factors, resulting in a more comprehensive argument.

5.     The sample size included in the study is relatively small, which is indeed a major limitation of the article.

Author Response

Dear reviewer 2,

Thank you for your comments and giving us the possibility to improve the manuscript. Below we have addressed all your comments – written in italic.

In fact, the incidence of malignant liver tumors in children, including in China, has been increasing year by year. This is partly due to improved diagnostic capabilities and the influence of factors such as food and the environment. Various treatment methods, including chemotherapy, as well as the use of multidisciplinary teams (MDT), have improved the effectiveness of treatment for malignant liver tumors in children. However, corresponding complications, such as renal toxicity and hearing impairment, are issues that clinicians must pay attention to. This article reveals the epidemiological characteristics of childhood malignant liver tumors in Denmark over the past few decades, which has certain clinical significance. The reviewer overall believes that revisions are necessary but maintains a positive attitude towards publication.

There are several issues with this article, as follows:

  1. In the introduction section, the third paragraph lacks content. The author is requested to provide additional information.

“all liver segments” has been added.

  1. The formatting of Table 1 needs to be standardized. It may be better to add parentheses for the quantity annotation in the first column.

Done

  1. The article includes statistical analysis of the incidence of hepatoblastoma and malignant liver tumors in different years (1985-1996, 1997-2008, 2009-2020, and 1985-2020). It may be worthwhile to further analyze and compare the long-term survival rates of hepatoblastoma and malignant liver tumors in these different years, providing more clinical guidance.

We chose two periods of diagnosis since there has not been major treatment protocol changes in the period of study, but rather an optimization of treatment administration methods and surgical techniques.

  1. Since age over 8 years is an important factor affecting prognosis, it would be beneficial to compare the cardiac, renal, and hearing complications of hepatoblastoma patients over 8 years old with those of the overall population of children with malignant liver tumors. The author can further conduct statistical analysis to identify any differences in risk factors, resulting in a more comprehensive argument.

Only 5 patients with hepatoblastoma were over 8 year of age at diagnosis, and only 2 are alive. Too few for statistical analysis.

  1. The sample size included in the study is relatively small, which is indeed a major limitation of the article.

Our study is population-based but in a small country

Reviewer 3 Report

This study retrospectively summarises the situation of malignant paediatric liver tumours in Denmark. It supplies more original data for studying liver cancers in children. 

Here are two minor issues/opinions:

1. Its English needs double-checking (I'm not a native English speaker), e.g.

line 28: outcomes (are) seldomly (seldom) published.

line 36: The survival after

line 40-41: Demark through (over) 35 years

line 46-47: "nine patients" should have more than two diseases, so "an" and "a" are not the best

line 53: "after the year 2000 and" might be better than "after year 2000 and"

2. In the discussion section, it might be better to add "4.2. Comparison with other studies." data into the result section: to be listed in suitable tables.

see above.

It's in good English, but not pleasant to read.

Author Response

Dear reviewer 3,

Thank you for your comments and giving us the possibility to improve the manuscript. Below we have addressed all your comments – written in italic.

This study retrospectively summarises the situation of malignant paediatric liver tumours in Denmark. It supplies more original data for studying liver cancers in children. 

Here are two minor issues/opinions:

  1. Its English needs double-checking (I'm not a native English speaker), e.g.

line 28: outcomes (are) seldomly (seldom) published. Done

line 36: The survival after . Done

line 40-41: Demark through (over) 35 years . Done

line 46-47: "nine patients" should have more than two diseases, so "an" and "a" are not the best . Done

line 53:"after the year 2000 and" might be better than "after year 2000 and" Done

  1. In the discussion section, it might be better to add "4.2. Comparison with other studies." data into the result section: to be listed in suitable tables.

We agree and we have tried to build a table, but the 4 “other studies” bring different information which will be lost in a table.